# Gill Oxidative Stress Protection through the Use of Phytogenics and Galactomannan Oligosaccharides as Functional Additives in Practical Diets for European Sea Bass (*Dicentrarchus labrax*) Juveniles

**DOI:** 10.3390/ani12233332

**Published:** 2022-11-28

**Authors:** Antonio Serradell, Daniel Montero, Álvaro Fernández-Montero, Genciana Terova, Alex Makol, Victoria Valdenegro, Félix Acosta, María Soledad Izquierdo, Silvia Torrecillas

**Affiliations:** 1Grupo de Investigación en Acuicultura (GIA), IU-ECOAQUA, Universidad de las Palmas de Gran Canaria, 35200 Las Palmas, Spain; 2Department of Pathobiology, School of Veterinary Medicine, University of Pennsylvania, Philadelphia, PA 19104, USA; 3Department of Biotechnology and Life Sciences, University of Insubria, 2-21100 Varese, Italy; 4Global Solution Aquaculture Unit, Delacon Biotechnik Gmbh, 24-4209 Engerwitzdorf, Austria; 5Biomar A/S, Global RD Health, BioMar AS, 2993 Trondheim, Norway

**Keywords:** European sea bass, functional diets, galactomannan–oligosaccharides, gill relative gene expression, low-FM/FO diets, oxidative stress, phytogenics

## Abstract

**Simple Summary:**

Under intensive aquaculture conditions, fish are subjected to a wide variety of stressors, making fish prone to suffering chronic stress and impairing fish growth performance and immune response. Fishmeal (FM) and fish oil (FO) replacement by raw terrestrial materials may induce nutritional imbalances, leading to a chronic stress status and oxidative stress processes. The functional ingredients have been profiled as suitable candidates to face these negative side effects by reinforcing fish immune response, attenuating fish stress response and reducing fish oxidative stress. The present study evaluates the effects of two different functional ingredients, plant origin galactomannan-oligosaccharides (GMOS) and a mixture of garlic and labiate plant essential oils (PHYTO), as potential boosters of gill endogenous antioxidant capacity in European sea-bass (*Dicentrarchus labrax)* juveniles fed low-FM/FO-based diets. After a confinement stress challenge (C challenge) or confinement combined with an in vivo infection with the pathogen *Vibrio anguillarum* (CI challenge), the functional ingredients induced a controlled pro-inflammatory response against the stressor. The functional ingredients attenuated fish stress response, leading to a stable energy metabolism and an ameliorated antioxidant status, altogether indicating the potential of both functional additives to reduce the associated negative effects of stress in European sea bass fed a low-FM/FO diet.

**Abstract:**

The aim of the present study is to evaluate the potential of two functional additives as gill endogenous antioxidant capacity boosters in European sea-bass juveniles fed low-FM/FO diets when challenged against physical and biological stressors. For that purpose, two isoenergetic and isonitrogenous diets with low FM (10%) and FO (6%) contents were supplemented with 5000 ppm plant-derived galactomannan–oligosaccharides (GMOS) or 200 ppm of a mixture of garlic and labiate plant essential oils (PHYTO). A control diet was void from supplementation. Fish were fed the experimental diet for nine weeks and subjected to a confinement stress challenge (C challenge) or a confinement stress challenge combined with an exposure to the pathogen *Vibrio anguillarum* (CI challenge). Both GMOS and PHYTO diets attenuated fish stress response, inducing lower circulating plasma cortisol and down-regulating *nfκβ2* and *gr* relative gene-expression levels in the gill. This attenuated stress response was associated with a minor energetic metabolism response in relation to the down-regulation of *nd5* and *coxi* gene expression.

## 1. Introduction

Fish reared under intensive aquaculture conditions are subjected to a wide variety of stressors. Between them, the nutritional imbalances may induce a chronic stress status [1,2,3] compromising fish growth performance and impairing fish immune response and tissue integrity [4,5,6,7,8].

Fish gills have essential functions for fish physiological balance, gas exchange, hydro–mineral balance [9], and immune response [10]. As gills interact directly with the external environment, they are the first barrier of protection against external agents such as pathogens and chemicals, acquiring a significant importance in fish development and disease resistance [10,11]. One of the main cell types composing gill epithelia are mitochondria-rich cells (MRCs), which are involved in gas exchange, ion transport, and blood acid–base balance regulation [9].

As a direct consequence of a stressful process, cortisol will target gills, increasing oxidative phosphorylation to ensure the energy availability to conduct all the physiological changes required to cope with the stress processes and an up-regulating Na^+^/K^+^ ATPase pump activity to maintain tissue hydro–mineral balance and functioning [12]. Cortisol effects are mediated through glucocorticoid receptors (GR), which reside in the cytoplasm complexed with the co-chaperone heat-shock proteins heat-shock protein 70 (HSP70) and heat-shock protein 90 (HSP90) [13]. Cortisol binds to the GR, inducing the dissociation of the chaperon proteins; then, cortisol–GR complex translocates into the nucleus to regulate gene expression of different stress-responsive factors, such as the pro-inflammatory nuclear factor *κβ* (*nfκβ2*). The NF*κβ* protein is one of the most important mediators of inflammatory response, which can be activated by different extracellular stimuli, such as pro-inflammatory cytokines [14,15], reactive oxygen species (ROS) [15,16], pathogen-associated molecular patterns (PAMPs) [17,18], and acute stress events [19]. In parallel, the GR can bind to the BCl-2 receptor in the mitochondrial membrane, inducing an increase in the oxidative phosphorylation rate by the mitochondrial electron-transport chain (ETC), generating energy to supply the adenosine triphosphate (ATP) synthase to produce ATP [20]. However, not all the electrons in the ETC are transferred to the final acceptor, generating an electron leak, which leads to the formation of reactive oxygen species (ROS)—namely, superoxide-radical (O_2_^−^) formation. Superoxide radicals are transformed by superoxide dismutase (SOD) to hydrogen peroxide (H_2_O_2_), which diffuses to the cytoplasm to be detoxified by glutathione peroxidase (GPX) and catalase (CAT) [21,22]. 

Oxidative stress results from an imbalance between ROS production and its neutralization by the antioxidant-defense system. It leads to the oxidation of essential biomolecules such as proteins and lipids, DNA damage, and the impairment of mitochondrial activity, causing cell death [21]. Insufficient ATP production will also impair Na^+^/K^+^ ATPase activity, causing hydro–mineral imbalances [23,24].

Functional ingredients have been profiled as suitable tools to face these harmful side effects, boosting fish health and promoting fish welfare, reinforcing fish immune response [25,26,27,28], attenuating fish stress response [28,29,30,31], and reducing oxidative stress processes [32,33,34]. This is of remarkable interest in European sea bass, which is an especially susceptible fish species to stress processes [35,36,37]. Among the functional ingredients, phytogenic feed additives (PFAs) are plant-derived bioactive compounds such as flavonoids, mucilages, and tannins with antioxidant properties [38,39,40]. In previous studies, dietary supplementation with a mixture of garlic and labiate plant essential oils attenuated European sea-bass juveniles’ stress response, with the fish fed the supplemented diets presenting lower circulating cortisol levels in comparison to fish fed a control diet [30,41,42]. In addition, PFA supplementation enhanced fish gut-mucosal health, reducing pre-ileorectal valve-segment goblet-cell size as compared to fish fed the control diet [41]. Another example of plant-derived functional ingredients are prebiotics, which are indigestible fibers with the ability to enhance host health by selectively stimulating the growth and activity of a limited number of intestinal bacterial species [28,43,44,45,46]. Between them, galactomannan–oligosaccharides (GMOS) have demonstrated in previously reported studies to increase host antioxidant capacity, modulate gut microbiota, and promote gut health in this fish species [41,42,47].

A scarce number of studies have investigated the effects of functional ingredients to offset the negative effects derived from low-FM/FO diet formulation and especially in fish subjected to stress processes. Thus, the aim of this study is to evaluate the effects of functional additives (PFAs and GMOS) as potential boosters of the gill endogenous antioxidant capacity of European sea-bass juveniles fed low-fish meal (FM)/fish oil (FO)-based diets when challenged against physical and biological stressors. 

## 2. Materials and Methods

### 2.1. Experimental Diets

Three isonitrogenous and isoenergetic low-fishmeal- and -fish oil-based diets (10% FM/6% FO) were formulated and produced by Biomar (Brande, Denmark), covering all the nutritional requirements for European sea bass. The control diet was void of functional ingredients (control diet), the GMOS diet was supplemented with 5000 ppm plant-derived galactomannan–oligosaccharides, and the PHYTO diet was supplemented with 200 ppm of a mixture of garlic and labiate plant essential oils. Functional additives were supplemented according to the producer’s commercial recommendations (Delacon Biotechnik GmbH, Engerwitzdorf, Austria). To ensure product stability, GMOS was included in the mix pre-extrusion process and replaced standard carbohydrates. PHYTO additive was included in the post-extrusion process by vacuum coating and homogenized with dietary oil. The stability of the used PFAs was checked prior to diet production and at the beginning of the feeding trial. The ingredients used in the diets and their proximate composition are detailed in Table 1, below [43].

### 2.2. Feeding Trial

This experiment is part of a series of experiments belonging to the project PROINMUNOIL PLUS, funded by the Spanish Ministry of Economy, Industry and Competitiveness. All the experiments were conducted in the facilities of the Parque Científico-Tecnológico Marino (PCTM) of the University of Las Palmas de Gran Canaria (ULPGC) (Canary Islands, Spain). A total of 675 European sea-bass juveniles from a local farm (Aquanaria, Castillo del Romeral, Gran Canaria, Canary Islands, Spain) were transferred and acclimatized for 4 weeks to the PCTM facility’s water conditions (6.1−6.6 ppm dissolved O_2_, 18.2–20 °C, 36 ppm salinity) under a natural photoperiod (12L:12D). After acclimation, the fish were randomly distributed in 9 fiberglass tanks of 500 L (75 fish/tank), having an initial mean weight of 23.5 ± 0.8 g. Each experimental group was triplicated (3 tanks/diet) and the experimental fish were fed 6 days a week, 3 times a day until apparent satiation for 9 weeks. Feed intake was monitored daily, and growth and feed efficiency were calculated at the end of the experimental period. The Bioethical Committee of the University of Las Palmas de Gran Canaria approved all the protocols used in the present study (approval no. OEBA_ULPGC_14/2020).

At the end of the feeding trial (t = 0 h sampling point), three fish per tank (*n* = 9 fish per dietary treatment) were used to obtain blood samples in order to evaluate fish plasma circulating cortisol and glucose levels as stress indicators (Figure 1).

### 2.3. Stress Challenge

After 9 weeks of the feeding experiment, a total of 90 fish per dietary treatment were transferred to the Marine Biosecurity (MBS) facilities of the PCTM-ULPGC (Taliarte, Canary Islands, Spain) and exposed to a stress challenge. Forty-five fish per dietary treatment were subjected to a confinement stress challenge (C challenge), consisting of a culture-density increase [48] (final stress challenge density = 35 kg/m^3^) by confinement in submerged nets (15 fish/net, 3 nets/dietary treatment). The other 45 fish per dietary treatment were exposed to the same confinement stress challenge combined with an in vivo exposure to *Vibrio anguillarum* (CI challenge) (10^5^ cfu/mL per fish, strain 7507, isolated from a clinical outbreak in Canary Islands) via intestinal inoculation as described before for *V. alginolyticus* [49]. The nets were placed in 6 fiberglass cylindroconical 500 L tanks on a RAS system supplied with filtered seawater at temperature of 22 °C, with 3 tanks for the C challenge and 3 tanks for the CI challenge. The fish were fed daily 3 times per day until apparent satiation during the entire stress challenge.

At 2 h, 24 h, and 168 h a whole net per dietary treatment and stress challenge was sampled to obtain blood for stress-indicator analysis (*n* = 9 fish per dietary treatment and stress challenge) and gill samples (*n* = 3 fish per dietary treatment and stress challenge) for stress and antioxidant response-related relative gene-expression analysis (Figure 1).

### 2.4. Sampling Methodology

In order to obtain blood samples, the fish were anesthetized with clove oil 0.02 mL/L (Guinama S.L; Pobla de Vallbona, Valencia, Spain; Ref. Mg83168) diluted in alcohol 100% (1:2). Afterwards, blood samples were obtained by caudal-sinus puncture with 1 mL syringes. Blood was stored in 1.5 mL Eppendorf tubes coated with heparin to avoid blood coagulation. Immediately, the blood was centrifuged at 3000× *g* at 4 °C for 5 minutes. The obtained plasma samples were rapidly kept at −80 °C until plasmatic-cortisol and glucose-concentration analysis. Plasmatic-cortisol concentration was determined using the assay kit Access Cortisol ref 33600, ©2010 Beckman Coulter, Inc., by the external laboratory AnimaLab (Las Palmas de Gran Canaria, Gran Canaria, Canary Island, Spain). Circulating plasma-glucose concentration was determined using the hexokinase method for in vitro diagnosis. The assay was performed using glucose-reactive OSR6521 from Beckman Coulter AU, employable on AU2700^®^ and AU5400^®^ chemistry analyzers (Beckman Coulter AU, PN B06960AA; March 2012).

Gill samples for relative gene expression were obtained after fish euthanasia by a clove-oil overdose of 0.5 mL/L (Guinama S.L; Spain, Ref. Mg83168) diluted in alcohol 100% (1:2). The second holobranch on the left side of the fish was excised with sterile dissection material and placed individually in 2 mL Eppendorf tubes filled with RNA later (for later 1.4 L RNA preparation: dilute in 1 L deionized water 650 g ammonium sulphate, 7.4 g sodium citrate dihydrate, 7.4 g EDTA disodium salt, and 200–500 µL concentrated sulfuric acid; final pH 5.2) for 24 h. Afterwards, the RNA later was removed and samples were frozen at −80 °C until gene-expression analysis.

### 2.5. RNA Extraction and Real-Time PCR Analysis

To perform the real-time PCR analysis, total gill mRNA (ng/μL) was extracted using TRI reagent (Sigma-Aldrich, St. Louis, MO, USA) from an RNeasy Minikit from Qiagen. An iScript^TM^ cDNA synthesis kit (Bio-Rad Hercules, California) was employed to perform the reverse transcriptions to obtain cDNA in a 20 μL reaction containing 1 μL of total mRNA. 

The real-time PCR analysis was performed with an iCycler with an optical module (Bio-Rad Hercules, CA, USA) in a final volume of 20 μL containing 10 μL of iQTM-SYBER Green Supermix (Bio-Rad Hercules, CA, USA), 5 μL of free-nuclease water, 3 μL of cDNA (1:10 dilution), and 1 μL of forward and reverse primers. The target genes were nuclear factor kappa beta (*nfκβ2*), hypoxia inducible factor 1 α (*hif-1α*), glucocorticoid receptors (*gr*), NADH dehydrogenase subunit 5 (*nd5*), cytochrome oxidase c subunit 1 (*coxi*), superoxide dismutase (*sod*), catalase (*cat*), glutathione peroxidase (*gpx*), tight cell-junction occludins (*ocln*), zonula occludens-1 (*zo-1*), heat-shock protein 70 (*hsp70*), heat-shock protein 90 (*hsp90*), and Na^+^/K^+^ ATPase subunit *α*1a (*NKA α*1a). Specific primer sequences and accession numbers for each target gene assayed are reported in Table 2. The real-time running conditions were 95 °C, 1 min followed by 40 cycles at 95 °C for 10 s and annealing temperature for 30 s (Table 2). All reactions were performed in duplicate for each template cDNA and a blank control containing nuclease-free water instead of cDNA was included in each assay as a negative control. Three constitutive genes were tested: *α*-tubuline (*α-tub*)*,* eukaryotic translation elongation factor 1 *α*1 (*eEF1α1*), and β-actin *(β-act*). After applying the CFX Maestro^TM^ Software selection tool (CFX Maestro™ Software User Guide Version 1.1, Biorad), the *α-tub* was selected as the most stable and amplification-efficient reference gene. Relative gene-expression levels were calculated using the 2^−ΔΔCt^ method [50], using *α-tubulin* as housekeeping gene. 

The gene expression was calculated relative to the transcript levels of 2 h post confinement stress challenge (C challenge) of fish fed the control diet.

### 2.6. Statistical Analyses

All the analyses were performed with R Project for Statistical Computing. Means and standard deviations (SD) were calculated for each parameter measured. For each sampling point, a two-way ANOVA analysis was performed to evaluate the effects of each dietary treatment on fish circulating plasma cortisol and glucose concentrations and gill relative gene expression in response to the different stress challenges. All data analyzed were tested for normality and homogeneity. When data did not accomplish homogeneity, the alpha-value was reduced to 0.01 in the analyses. When significant differences were obtained, a Tukey post-hoc test was performed for multiple-means comparison.

## 3. Results

As reported in our previous studies [44], fish grew properly during the feeding trial with no significant effects on fish growth performance associated with the use of the functional diets. After the nine-week feeding trial, the fish presented a mean increase of 2.6× body weight, representing a relative growth (%) of 158.8 ± 16.3. During the feeding trial, no mortality was registered in any of the specific dietary treatments.

Functional-ingredient dietary inclusion did not induce any differences on fish basal (t = 0 h) stress parameters, with values ranging from 4.67 to 5.82 ng/mL for circulating plasma cortisol and from 67.43 to 67.71 mg/dL for circulating plasma glucose (Table 3). At 2 h after crowding stress a generalized increase (*p* < 0.05) in circulating plasma-cortisol concentration was observed, with significantly higher (*p* < 0.05) values in those fish fed the control diet. In the early hours after C challenge (2 h), all the dietary treatments presented an increase in circulating cortisol levels, especially those fish fed the control diet, with significantly higher (*p* < 0.05) levels than fish fed GMOS and PHYTO diets (*p* < 0.05). At 2 h after CI challenge, a similar trend was observed, with fish fed GMOS presenting lower (*p* < 0.05) cortisol levels than fish fed the control diet. On the contrary, at 24 h after CI challenge fish fed GMOS and PHYTO diets presented significantly higher (*p* < 0.05) circulating plasma-cortisol levels than those fed the control diet (Table 3). At the end of the CI challenge (t = 168 h), fish fed the PHYTO diet presented higher (*p* < 0.05) levels of circulating plasma cortisol than fish fed the control diet. Regarding circulating plasma-glucose concentrations, the use of functional additives did not induce significant differences in fish pattern of response against crowding stress (C challenge). Meanwhile, at 2 h and 168 h after the CI challenge, fish fed GMS and PHYTO diets presented significantly higher (*p* < 0.05) plasma-glucose levels than fish fed the control diet (Table 3).

At the end of the stress challenge, fish subjected to the confinement (C challenge) did not present mortality regardless of dietary treatment. Nevertheless, the combination of both confinement and the pathogen gut inoculation (CI challenge) resulted in a relative percentage of survival (RPS = [1 – (%) surviving fish fed functional diet/(%) surviving fish fed control diet] × 100) of 47% and 33% in fish fed the PHYTO and GMOS diets, respectively (Figure 2), compared to fish fed the control diet [45]. 

When fish were subjected to the stress challenge, two hours after confinement (C challenge) those fed the control diet presented significantly higher (*p* < 0.05) gill-transcript levels of *nfκβ2*, *hif-1α*, *gr*, *nd5*, *coxi*, *sod*, *cat*, *hsp70*, *hsp90*, and *NKA α1a* genes than those fed GMOS and PHYTO diets (Table 4, Figure 3). No differences were found among fish fed the different dietary treatments and subjected to C challenge for *gpx*, *ocln*, and *zo-1* relative gene expression. Similarly, two hours after the CI challenge, fish fed the control diet presented a significant (*p* < 0.05) up-regulation of *nfκβ2*, *hif-1α*, *gr*, *sod*, *gpx*, *NKA α1a*, and *hsp90* gill gene expression compared to gills of fish fed GMOS and PHYTO diets. At this sampling point, fish fed the control diet and subjected to the CI challenge presented significantly lower (*p* < 0.05) *nd5* and *coxi* gill transcript levels than those fed the same diet but subjected to the C challenge.

At 24 h after confinement and in relation to the gene-expression levels observed after 2 h of confinement, fish fed the control diet presented a significant down-regulation (*p* < 0.05) of *nfκβ2*, *gr*, *nd5*, *coxi*, and *hsp70* and an up-regulation (*p* < 0.05) of *zo-1* gill relative gene expression compared to the initial transcript levels at 2 h after confinement (Figure 4, Table 5). On the contrary, fish fed the GMOS diet presented an up-regulation (*p* < 0.05) of *nfκβ2*, *hif-1α*, *gr*, *sod*, *cat*, *hsp70*, and *hsp90* gill relative gene expression and fish fed the PHYTO diet presented an up-regulation of *hif-1α* compared to the previous sampling point at 2 h. 

For fish confined and infected after 24 h post challenge the pattern of response for all the dietary treatments resulted in a down-regulation (*p* < 0.05) of *ocln* gill relative gene expression compared to the previous sampling point, whereas *coxi* and *cat* transcript levels presented a significant up-regulation (*p* < 0.05) compared to the initial transcript levels. Besides, fish fed the control and PHYTO diets presented an up-regulation (*p* < 0.05) of *sod* gill relative gene expression compared to 2 h post CI challenge. Fish fed the GMOS diet, and in relation to 2 h post CI challenge, presented a significant up-regulation (*p* < 0.05) of *hif-1α* gill transcript levels, whereas fish fed the PHYTO diet presented an up-regulation (*p* < 0.05) of *nfκβ2*, *gr*, and *NKA α1a* gill gene expression compared to 2 h post CI challenge.

Within the 24 h sampling point for the C challenge, fish fed the GMOS diet presented higher (*p* < 0.05) gill gene-transcript levels of *nfκβ2*, *sod*, *gpx*, and *hsp90* than those fed the control diet, as well as higher (*p* < 0.05) *sod* gill relative gene-expression values than those fed the PHTYO diet. Similarly, fish fed the PHYTO diet presented higher (*p* < 0.05) gill relative gene expression of *ocln* than those fed the control diet. On the other hand, in fish subjected to the CI challenge after 24 h of stress challenge, the diet did not induce significant differences in the gill relative gene expression of the target genes, despite both stress challenges differing between them in the gill relative gene-expression patterns presented. At 24 h post CI challenge, those fish fed the control diet presented higher (*p* < 0.05) *sod* and *hsp70* gene-expression levels than those fed the same dietary treatment but subjected to the C challenge. On the contrary, the CI challenge induced a significant down-regulation (*p* < 0.05) of *sod* gill transcript levels in fish fed the GMOS diet and subjected to the C challenge. Regardless of the dietary treatment, the CI challenge induced a significant down-regulation (*p* < 0.05) of *zo-1* relative gene expression compared to C challenge after 24 h post challenge.

At the end of the confinement stress challenge (168 h after C challenge), fish fed the GMOS diet presented a significant down-regulation (*p* < 0.05) of *sod* and *cat* gill transcript levels (Table 6, Figure 5) down to the levels observed at 2 h post challenge, as presented by fish fed the PHYTO diet for *zo-1* gill relative gene expression. Fish fed the control diet and confined presented a significant down-regulation (*p* < 0.05) of *NKA α1a* gill relative gene expression compared to the previous sampling points at 2 h and 24 h post C challenge. In the case of the CI challenge, fish fed GMOS presented a significant down-regulation (*p* < 0.05) of sod gill gene-expression levels down to the initial levels. 

Within this last sampling point (168h), fish from the C challenge and fed GMOS diet presented higher (*p* < 0.05) *NKA α1a* gill relative gene expression than those fed the control diet, whereas fish fed the PHYTO diet presented lower (*p* < 0.05) gill gene expression *ocln* than those fed the control diet. Regarding the CI challenge, fish fed the control diet presented higher (*p* < 0.05) gill *sod* transcript levels than fish fed the GMOS and PHYTO diets, whereas fish fed the control and PHYTO diet presented higher (*p* < 0.05) *gr* gill relative gene-expression values than those presented by fish fed the GMOS diet. On the contrary, fish fed the GMOS diet presented higher (*p* < 0.05) *ocln* and *zo-1* gill gene-expression levels than fish fed the control and PHYTO diets, respectively.

At 168 h after CI challenge, the infection itself induced significantly higher (*p* < 0.05) *sod*, *NKA α1a*, and *hsp70* gill gene-transcript levels, whereas it down-regulated (*p* < 0.05) *ocln* gill gene expression. In the case of the fish fed the GMOS diet, the CI challenge significantly increased (*p* < 0.05) *ocln* and *zo-1* gill relative gene expression and reduced (*p* < 0.05) *hsp90* in comparison to fish subjected to the C challenge. No effects were detected for fish fed the PHYTO diet when comparing the C and CI challenges at the end of the stress trial.

## 4. Discussion

Stress induces a physiological response to reestablish fish homeostasis, which is orchestrated by the release of cortisol into the bloodstream. As expected, in the present study an increased concentration of fish circulating plasma concentration was observed in the few hours after the initiation of both stress challenges and regardless of the dietary treatment. However, both GMOS and PHYTO diets attenuated fish stress response, with supplemented fish presenting lower circulating-cortisol levels than fish fed the control diet. 

In response to the stress process, the organism undergoes an alarm status in which the energetic resources are rearranged in order to cope with the surplus of activity. In this sense, gills play a fundamental role in energy supply with the activation of ATP synthesis [20,51]. In the present study, and in agreement with the attenuated stress response observed in cortisol patterns for supplemented fish, at 2 h after both the C and CI challenge, fish fed the GMOS and PHYTO diets presented lower *gr*, *nd5*, and *coxi* gill relative gene expression than fish fed the control diet. This may suggest a lower responsiveness of fish fed functional diets against the stressor, with a lower activity of the ETC and thus lower energetic requirements to cope with the stress process [51]. Nevertheless, fish fed the GMOS and PHYTO diets presented an up-regulation of *gr* relative gene expression at 24 h after the stress challenges. This delayed increase in *gr* gene expression in fish fed the supplemented diets could be understood as a mechanism to restore the GR protein content after exposure to glucocorticoids. Vijayan t al. (2013) reported on an in vitro experiment using hepatocytes, in which down-regulated *gr* gene expression was found in treatments presenting the higher GR protein contents [52]. 

Considering that the ETC is one of the major sources of endogenous ROS [20,34,53], an increased aerobic-metabolism rate in response to a stress process may induce elevated ROS production, leading to oxidative stress [21]. In the present study, at 2 h after the C challenge, fish fed the functional diets presented lower *sod* and *cat* gill gene-expression levels. A lower activation of the endogenous antioxidant defenses could suggest an attenuated stress response, leading to a lower production of mitochondrial ROS and thus an attenuated response against the stress processes. Although both functional diets reduced fish-stress and oxidative-stress response, each functional diet induced a different antioxidant response against the confinement stress challenge. In fact, along with the stress challenge, fish fed PHYTO did not presented an activation of endogenous antioxidant machinery gene expression. This lack of response in the endogenous antioxidant defenses could be present and associated with the antioxidant properties of the phytogenic compounds, making them capable of inhibiting ROS formation and quenching them once they are formed [54,55,56]. Indeed, our previous studies in this mucosal tissue indicated that both functional additives can reduce gill oxidative-stress status in basal conditions after dietary supplementation [42]. Between the different plant-origin biomolecules, the flavonoids are polyphenolic compounds with high antioxidant properties and can be found in a wide spectrum of plant extracts as garlic oil and labiate plant extracts, such as origanum [57] and thyme [58]. On the other hand, fish fed the GMOS diet presented an increase in *sod* and *cat* gill gene expression at 24 h after the C challenge. In addition, fish fed the GMOS diet presented higher *gpx* gill transcript levels than fish fed the other dietary treatments. Interestingly, our previous studies indicated a similar delayed pattern of response against the stressor in other mucosal tissues, inducing GMOS supplementation as a controlled and prolonged intestinal-mucus secretion in response to the CI challenge, reducing gut bacterial-translocation rates and thus increasing pathogen resistance and survival rates [41]. Similarly, in the present study both functional diets also attenuated fish antioxidant-related gill gene response against the CI challenge, with fish fed the functional diets presenting lower values of *sod* and *gpx* gene expression than fish fed the control diet in the early hours after bacterial gut inoculation. Nevertheless, 24 h after the CI challenge the functional-diet supplementation induced an up-regulation of *sod*, *cat*, and *gpx* gill gene expression. This increase in the expression of antioxidant-related genes might suggest a response against the stress associated with the pathogen gut inoculation and with an organism arrangement to cope with future infection. In particular, dietary PHYTO increased fish *gpx* gene expression and kept it up-regulated throughout the entire stress challenge. Similarly, Mansour et al. (2020) observed a higher antioxidant capacity in gilthead sea-bream (*Sparus aurata*) gills and skin, with an increased gene expression of *sod* and *cat* after feeding diets supplemented with *Moringa oleifera* leaf extracts [59].

The gills, skin, and intestine are the first barrier interacting with the external environment, playing a fundamental role in maintaining tissue structure and integrity, regulating solute trafficking across the gill epithelium and thus facilitating or limiting paracellular-ion movement [60]. For this reason, cell-junction complexes play a fundamental role in maintaining gill-epithelium integrity and functioning. Damage to fish-gill structural integrity like that originated by oxidative-stress processes may lead to degenerative processes such as gas-exchange disturbances [61] and impairments to immune functions [62]. In our previous studies, Torrecillas et al. (2021) observed the ability of GMOS and PHYTO diets to induce higher gill gene expression of *zo-1* and *ocln*, respectively, in basal conditions [42]. In the present study, the use of different dietary treatments did not affect fish gill *ocln* and *zo-1* relative gene expression in the early hours after both stress challenges. However, 24 h after confinement, fish fed all the dietary treatments presented an increase in *zo-1* gill gene expression despite it only being significant in fish fed the control diet, with fish fed PHYTO diet at 24 h post confinement presenting the highest expression levels. In concordance with these results, Zhao et al. (2020) described an up-regulated gene expression of *ocln* and *zo-1* genes in the gills of *Ctenopharyngodon idella*-fed diets supplemented with *Allium monoglicum* Regel flavonoids (AMRF), alleviating the oxidative stress and toxicity derived from chromium exposure [56]. In the same way, Trujillo et al. (2015), described the ability of curcumin to prevent cisplatin-induced fibrosis and decreased tight-junction proteins in rat kidney [63]. These protective effects of phytogenic feed additives could be related to the ability of those plant-origin compounds to interact with MAPK receptors, preventing H_2_O_2_-induced tight-junction disruptions [64]. On the contrary, at 24 h after challenge when the fish were subjected to the CI challenge and at 24 h after stress all the dietary treatments presented a down-regulation of *ocln* gill gene expression. Moreover, the CI challenge induced a down-regulation of *zo-1* compared to the C challenge. Acute inflammatory processes are characterized by the hyper-permeabilization of tissues, allowing inflammatory mediators and immune cells to infiltrate the damaged tissues [65]. In this sense, a down-regulation of genes related to tight-junction structure maintenance could be related to a preparation process to facilitate the response against a future infection. The inflammatory response acquires a critical importance in the gills, considering the high number of permanent-resident lymphocytes and immune cells associated with the gill-associated lymphoid tissue (GIALT) [66]. In the present study, both the C and CI challenge induced an acute response of *nfκβ2* gill gene expression after stress challenge in fish fed the control diet, whereas fish fed the GMOS and PHYTO diets presented an attenuated pro-inflammatory response against the stressors, with the highest *nfκβ2* gill gene-expression levels at 24 h after being subjected to the C and CI challenge. Previous studies have reported on the ability of phytogenic compounds derived from oregano, curcumin, and thymol to modulate pro-inflammatory response in fish [67,68,69]. These compounds have been shown to be able to directly regulate the NFKB- and mitogen-activated protein kinase (MAPK)-signaling pathways, attenuating the inflammatory response [27]. In the case of GMOS, the mechanism that can modulate European sea-bass immune and stress response differs from the phytogenic compounds, as the animal does not directly harness the prebiotics. The by-products from prebiotic fermentation generated by the host microbiome may produce short-chain fatty acids (SCFAs), which can modulate fish innate immune response and inflammatory cells [70] by interacting with immune-cell pattern-recognition receptors [71]. Inflammatory processes are characterized by an increased leukocyte infiltration [5], which may lead to hypoxia conditions due to the high amount of O_2_ consumed by the increased phagocytic activity [72]. In response to the hypoxia, the *nfκβ2* triggers the activation of the *hif-1α,* inducing a metabolic swift into a glycolytic strategy, facilitating leukocyte survival in a hypoxic medium [73]. In the present study, fish fed the functional diets presented lower *hif-1α* gill gene-expression levels than those fed the control diet, supporting the idea of an attenuated pro-inflammatory response in the early hours after the stress. Nevertheless, at 24 h after stress challenge, and in parallel with an increased transcription of the *nfκβ2,* both functional diets induced an up-regulation of *hif-1α* gill gene expression. This could suggest a better protection of the immune-cell populations, leading to a better ability to cope with the deleterious effects derived from a prolonged inflammatory response against a stressor, which may also be related with the lower infection rates and higher survival observed in CI fish fed the functional diets [41]. Indeed, the fish fed the control diet presented no significant changes in *hif-1α* gill gene-expression levels regardless of the variations in the *nfκβ2* gene transcripts during the different stress challenges. A similar response was observed in previous reports, in which the same functional diets protected head-kidney leukocyte populations against apoptotic processes by attenuating head-kidney pro-inflammatory response and increasing *hif-1α* head-kidney relative gene expression after CI challenge [30].

Another mechanism promoting tissue integrity in response to a stressor is the activation of the heat-shock proteins, which are overexpressed to act as molecular chaperones associated with the GR avoiding protein denaturation, refolding denatured proteins and promoting misfolded-protein degradation [74,75,76]. In response to an acute stress process, the *hsp70* and *hsp90* gene expression is increased, activating the necessary mechanisms to respond to the stressor [77]. In the present study, in fish fed the control diet both C and CI challenges induced an overexpression of gill *hsp70* in the first hours after the challenge, followed by a strong down-regulation until the end of the stress challenges. Meanwhile, the fish fed the GMOS diet presented a delayed *hsp70* and *hsp90* gene-expression pattern, with the highest expression levels at 24 h after stress, indicating an attenuated response to the stress. In concordance, in previous studies dietary supplementation with fructo-oligosaccharides in blunt-snout bream (*Megalobrama amblycephala*) induced an increase in *hsp70* and *hsp90* at 24 hours after confinement stress [78].

Na^+^/K^+^ ATPase is an ATP-dependent transmembrane enzyme that plays a fundamental role in maintaining cell ionic homeostasis. This protein is highly represented in the gills and confers an important osmoregulatory role to the tissue [79]. In the present study, at 2 h after C and CI challenges, the fish fed the control diet presented a strong up-regulation of *NKA α1a* gill gene expression, indicating an acute response to the stressor. At the end of the confinement stress challenge, these fish presented a down-regulated gene expression of the *NKA α1a*. Alterations to cellular ionic balance may lead to the entrance of sodium (Na^+^) [23,80] and thus disturb the osmotic balance, leading to membrane ruptures [43]. When fish were subjected to the CI challenge, the control group presented the same pattern of response but the *NKA α1a* gene expression remained unchanged throughout the stress challenge, being highest at the end of the CI challenge. Meanwhile, and regardless of the stress challenge, fish fed the functional diets presented the higher values of relative gene expression of *NKA α1a* at 24 h after stress, indicating an attenuated response to the stressor. The fish fed the functional dies did not show a down-regulation of *NKA α1a*, which could suggest a more prolonged activity of the Na^+^/K^+^ ATPase and thus a better capacity to cope with the imbalances originated during the stress process, which in turn may also be related to the lower infection rates observed in supplemented fish.

## 5. Conclusions

In conclusion, both GMOS (5000 ppm) and PHYTO (200ppm) functional additives in 10% FM/6% FO diets induced a down-regulation of the *nf-κβ* relative gene expression in the gill during the stress challenge, leading to a controlled inflammatory response against the stressor. The functional diets attenuated fish stress response, leading to a stable energy metabolism and an ameliorated antioxidant status. Altogether, this indicates the potential of both functional additives to reduce the associated negative effects of stress in European sea bass fed a low-FM/FO diet. Owing to the diverse methods of action of the different functional additives analyzed in the present study, more experiments must be carried out to fully understand the potential effects on fish health and stress response.

## Figures and Tables

**Figure 1 animals-12-03332-f001:**
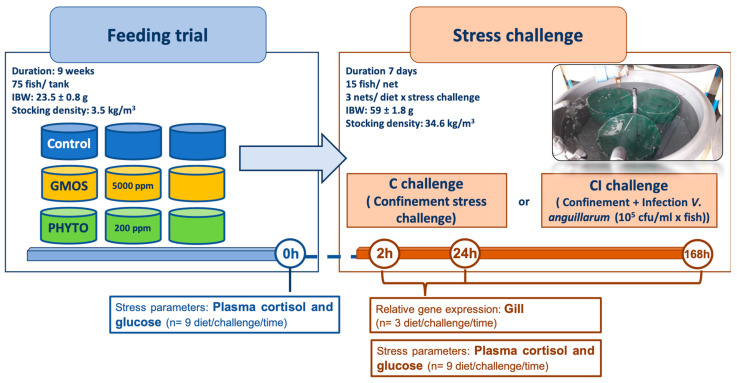
Experimental-design scheme. Nine-week feeding trial; three experimental treatments fed in triplicate (Control (no supplementation); GMOS (GMOS supplemented, 5000 ppm galactomannan-oligosaccharides); PHYTO (PHYTO supplemented, 200 ppm mixture of garlic and labiate plant essential oils); 3 times/day, 6 days/week until apparent satiation) (*n* = 75 fish/tank; initial body weight (IBW) = 23.5 ± 0.8 g). Seven-day stress challenge with two experimental treatments: C challenge (confinement stress challenge; 3 nets/dietary treatment, 15 fish/net, initial body weight (IBW) = 59 ± 1.8 g) or CI challenge (confinement stress challenge + intestinal infection with *V. anguillarum* (10^5^ cfu/mL × fish); 3 nets/dietary treatment, 15 fish/net, initial body weight (IBW) = 59 ± 1.8 g). Sampling points at t = 0 h, 2 h, 24 h, and 168 h after stress challenge: Stress parameters (blood plasma cortisol and glucose concentration, *n* = 9 diet/stress challenge/sampling point); relative gene expression (gill, *n* = 3 diet/stress challenge/sampling point).

**Figure 2 animals-12-03332-f002:**
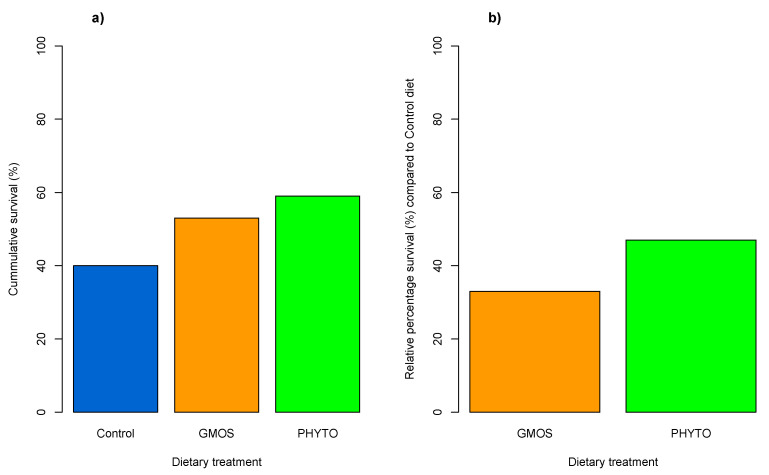
(**a**) Bar plot of cumulative survival (%) at the end of the CI challenge: control diet induced a 40% survival; GMOS diet (supplemented with 5000 ppm galactomannan–oligosaccharides) induced a 53% survival, PHYTO diet (supplemented with 200 ppm mixture of garlic and labiate plant essential oils) induced a 59% survival. (**b**) Bar plot of relative percentage of survival (RPS) (%) induced by GMOS and PHYTO diets in comparison to fish fed the control diet: GMOS diet (supplemented with 5000 ppm galactomannan–oligosaccharides) induced a 33% RPS; PHYTO diet (supplemented with 200 ppm mixture of garlic and labiate plant essential oils) induced a 47% RPS. Results previously reported in [41].

**Figure 3 animals-12-03332-f003:**
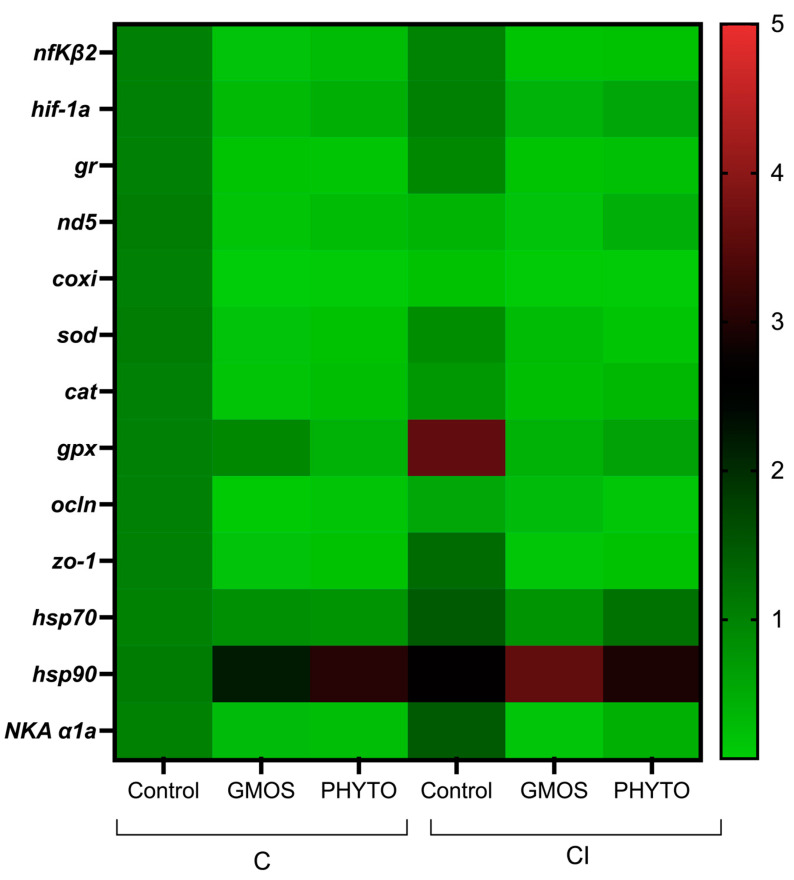
Heatmap of *Dicentrarchus labrax* gill relative gene expression at 2 h post stress challenge. Confinement stress challenge (C challenge). Confinement combined with infection with *Vibrio anguillarum* stress challenge (CI challenge). Control (control diet), GMOS (GMOS diet, 5000 ppm galactomannan–oligosaccharides), PHYTO (PHYTO diet, 200 ppm mixture of garlic and labiate plant essential oils). *n* = 3 samples/diet/challenge. Target genes: *nfκβ2*: nuclear factor kappa beta-2, *hif-1α:* hypoxia-inducible factor 1 alpha, *gr*: glucocorticoid receptor, *nd5*: NADH dehydrogenase subunit 5, *coxi*: cytochrome c oxidase subunit 1, *sod*: superoxide dismutase, *cat*: catalase, *gpx*: glutathione peroxidase, *zo-1*: zonula occludens-1, *ocln*: occludin, *hsp70*: heat-shock protein 70, *hsp90*: heat-shock protein 90, *NKA α1a*: Na^+^/K^+^ ATPase subunit *α*1a, *α-tubulin* (housekeeping).

**Figure 4 animals-12-03332-f004:**
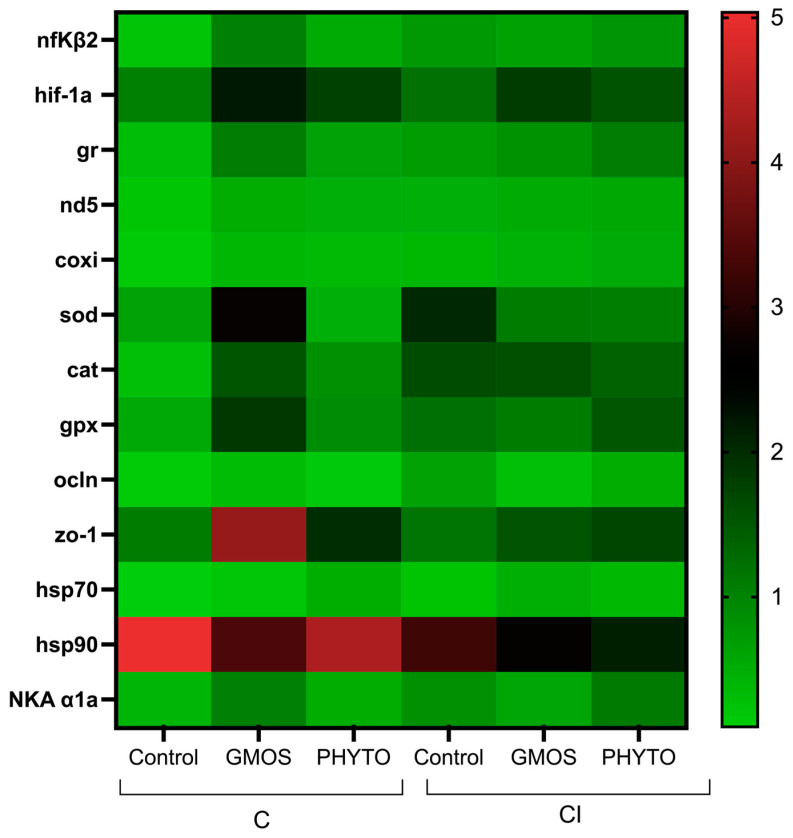
Heatmap of *Dicentrarchus labrax* gill relative gene expression at 24 h post stress challenge. Confinement stress challenge (C challenge). Confinement combined with infection with *Vibrio anguillarum* stress challenge (CI challenge). Control (control diet), GMOS (GMOS diet, 5000 ppm galactomannan–oligosaccharides), PHYTO (PHYTO diet, 200 ppm mixture of garlic and labiate plant essential oils). *n* = 3 samples/diet/challenge. Target genes: *nfκβ2*: nuclear factor kappa beta-2, *hif-1α:* hypoxia-inducible factor 1 alpha, *gr*: glucocorticoid receptor, *nd5*: NADH dehydrogenase subunit 5, *coxi*: cytochrome c oxidase subunit 1, *sod*: superoxide dismutase, *cat*: catalase, *gpx*: glutathione peroxidase, *zo-1*: zonula occludens-1, *ocln*: occludin, *hsp70*: heat-shock protein 70, *hsp90*: heat-shock protein 90, *NKA α1a*: Na^+^/K^+^ ATPase subunit *α*1a, *α-tubulin* (housekeeping).

**Figure 5 animals-12-03332-f005:**
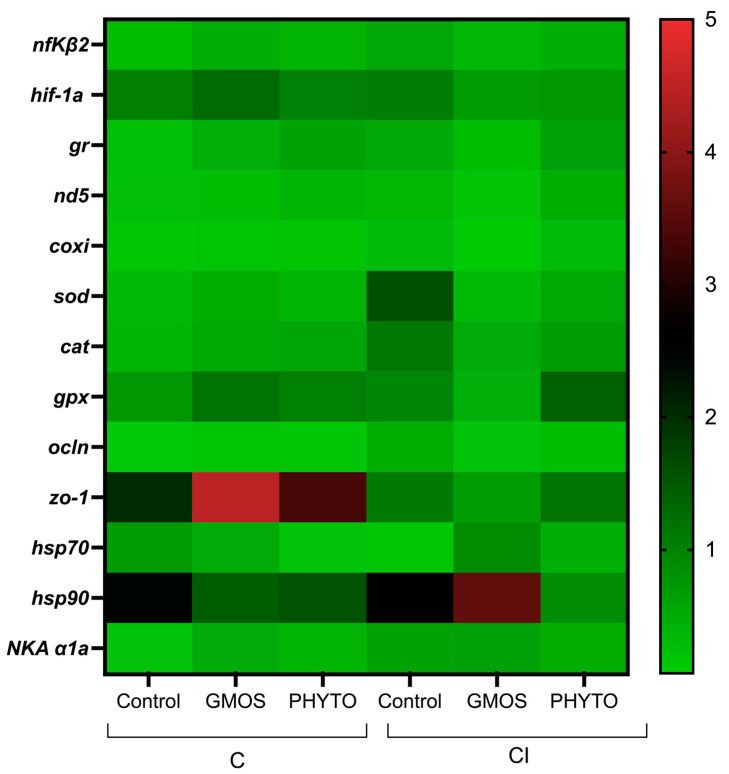
Heatmap of *Dicentrarchus labrax* gill relative gene expression at 168 h post stress challenge. Confinement stress challenge (C challenge). Confinement combined with infection with *Vibrio anguillarum* stress challenge (CI challenge). Control (control diet), GMOS (GMOS diet, 5000 ppm galactomannan–oligosaccharides), PHYTO (PHYTO diet, 200 ppm mixture of garlic and labiate plant essential oils). *n* = 3 samples/diet/challenge. Target genes: *nfκβ2*: nuclear factor kappa beta-2, *hif-1α:* hypoxia-inducible factor 1 alpha, *gr*: glucocorticoid receptor, *nd5*: NADH dehydrogenase subunit 5, *coxi*: cytochrome c oxidase subunit 1, *sod*: superoxide dismutase, *cat*: catalase, *gpx*: glutathione peroxidase, *zo-1*: zonula occludens-1, *ocln*: occludin, *hsp70*: heat-shock protein 70, *hsp90*: heat-shock protein 90, *NKA α1a*: Na^+^/K^+^ ATPase subunit *α*1a, *α-tubulin* (housekeeping).

**Table 1 animals-12-03332-t001:** Main ingredients and analyzed proximal composition of the experimental diets.

Ingredients	Diet (%)		
	Control	GMOS	PHYTO
Fishmeal ^1^	9.6	9.6	9.6
Soya protein concentrate	18.2	18.2	18.2
Soya meal	11.6	11.6	11.6
Corn gluten meal	24.1	24.1	24.1
Wheat	8.54	8.04	8.52
Wheat gluten	1.9	1.9	1.9
Guar meal	7.7	7.7	7.7
Rapeseed extracted	3.0	3.0	3.0
Fish oil ^2^	6.5	6.5	6.5
Rapeseed oil ^3^	5.2	5.2	5.2
Vitamin and mineral premix ^4^	3.6	3.6	3.6
Antioxidant ^5^	0.06	0.06	0.06
Galactomannan–oligosaccharides (GMOS) ^6^	0	0.5	0
Phytogenic ^7^	0	0	0.02
Proximate composition (% of dry matter)			
Crude lipids	19.91	20.44	20.47
Crude protein	49.30	49.27	49.76
Moisture	5.10	5.01	5.06
Ash	7.02	6.41	6.49

Dietary-ingredient composition and proximal composition expressed as % of dry weight. Control (Control diet), GMOS κGMOS diet, 5000 ppm galactomannan–oligosaccharides), PHYTO (PHYTO diet, 200 ppm mixture of garlic and labiate plant essential oils); ^1^ South American, Superprime 68%; ^2^ South American fish oil; ^3^ DLG AS, Denmark; ^4^ Vilomix, Denmark; ^5^ BAROX BECP, Ethoxyquin; ^6^ Delacon Biotechnik GmbH, Austria; ^7^ Delacon Biotechnik GmbH, Austria.

**Table 2 animals-12-03332-t002:** Primer sequences of the different genes analyzed and their RT-PCR conditions.

Gene	Accession Number	Primer	Nucleotide sequence 5′–3′	Annealing T (°C)
*nfΚβ2*	KM225790	Fw	CTGGAGGAAACTGGCGGAGAAGC	60
		Rv	CAGGTACAGGTGAGTCAGCGTCAC	
*hif-1α*	DQ171936	Fw	GACTTCAGCTGCCCTGATTC	60
		Rv	GGCTGGTTTATAGCGCTGAG	
*gr*	AY549305.1	Fw	GTGGGCCTACAAGACCAGAA	60
		Rv	CGGACGACTCTCCATACCTG	
*nd5*	KF857307	Fw	CCCGATTTCTGTGCCCTACTA	60
		Rv	AGGAAAGGAGTGCCTGTGA	
*coxi*	KF857308	Fw	ATACTTCACATCCGCAACCATAA	60
		Rv	AAGCCTCCGACTGTAAATAAGAA	
*sod*	FJ860004.1	Fw	CATGTTGGAGACCTGGGAGA	60
		Rv	TGAGCATCTTGTCCGTGATGT	
*cat*	FJ860003.1	Fw	TGGGACTTCTGGAGCCTGAG	60
		Rv	GCAAACCTCGATCGCTGAAC	
*gpx*	FM013606.1	Fw	AGTTCGTGCAGTTAATCCGGA	60
		Rv	GCTTAGCTGTCAGGTCGTAAAAC	
*zo-1*	MH321323.1	Fw	CGGCCTGCAGATGTTCCTAA	60
		Rv	GCTGAGGGAATTGGCTTTGA	
*ocln*	MH321322.1	Fw	GGACGAAGACGACAACAACGA	60
		Rv	CCATGGGAGAAAGCCTCTGA	
*hsp70*	AY423555.2	Fw	GGACATCAGCCAGAACAAGAGA	60
		Rv	GCTGGAGGACAGGGTTCTC	
*hsp90*	AY395632	Fw	GCTTCGAGGTCCTGTACATG	62.7
		Rv	GCCTTATCCTCCTCCATC	
*NKA α1a*	KP400258	Fw	AACCTCAGATGGCAAGGAGAAG	60
		Rv	GAGACTGGTACATTCAGGCGG	
*α-tub (hk)*	AY326429.1	Fw	AGGCTCATTGGCCAGATTGT	60
		Rv	CAACATTCAGGGCTCCATCA	
*eEF1α1*	XM_051391260.1	Fw	GTTGCTGCTGGTGTTGGTGAG	60
		Rv	GAAACACACTGCTGGAGGCTC	
*β-act*	AY148350.1	Fw	TCTTCCAGCCTTCCTTCCTC	60
		Rv	GATGTCAACGTCGCACTTCA	

Target genes: *nfκβ2*: nuclear factor kappa beta-2, *hif-1α:* hypoxia-inducible factor 1 alpha, *gr*: glucocorticoid receptor, *nd5*: NADH dehydrogenase subunit 5, *coxi*: cytochrome c oxidase subunit 1, *sod*: superoxide dismutase, *cat*: catalase, *gpx*: glutathione peroxidase, *zo-1*: zonula occludens-1, *ocln*: occludin, *hsp70*: heat-shock protein 70, *hsp90*: heat-shock protein 90, *NKA α1a*: Na^+^/K^+^ ATPase subunit *α*1a, *α-tub: α-tubulin* (housekeeping), *eEF1α1: Eukaryotic translation elongation factor 1 α1, β-act: β-actin*. Fw: forward, Rv: reverse.

**Table 3 animals-12-03332-t003:** Concentration of circulating plasma cortisol (ng/mL) and glucose (mg/dL) in European sea-bass juveniles.

	Confinement (C Challenge)			Confinement + Infection (CI Challenge)		
Plasma Cortisol (ng/mL)	Control	GMOS	PHYTO	Control	GMOS	PHYTO
0 h (basal)	5.82 ± 3.45	5.33 ± 7.06	4.67 ± 8.04	5.82 ± 3.45	5.33 ± 7.06	4.67 ± 8.04
2 h	321.83 ^a^ ± 171.51	270.86 ^b^ ± 87.28	307.00 ^b^ ± 53.93	611.29 ^a^ ± 185.62	254.29 ^b^ ± 121.57	374.50 ^ab^ ± 133.29
24 h	71.00 ± 46.67	22.20 ± 10.43	29.43 ± 10.13	47.80 ^a^ ± 38.79	145.33 ^b^ ± 68.55	77.50 ^b^ ± 38.28
168 h	16.67 ± 4.73	15.60 ± 11.50	100.43 ± 76.97	14.17 ^b^ ± 18.17	16.40 ^b^ ± 15.16	217.43 ^a^ ± 96.14
Plasma glucose (mg/dL)						
0 h (basal)	67.63 ± 16.78	67.43 ± 10.66	67.71 ± 10.95	67.63 ± 16.78	67.43 ± 10.66	67.71 ± 10.95
2 h	143.33 ± 69.61	156.60 ± 18.61	194.50 ± 16.26	236.00 ^a^ ± 55.48	131.33 ^b^ ± 48.58	158.00 ^b^ ± 22.63
24 h	77.20 ± 17.54	96.50 ± 22.02	95.80 ± 16.08	102.33 ± 28.38	75.33 ± 10.98	80.50 ± 7.78
168 h	75.20 ± 16.08	93.00 ± 20.17	124.00 ± 54.21	74.25 ± 7.46	87.00 ± 23.39	148.00 ± 26.87

Different letters denote significant differences among dietary treatments at each stress challenge (*p* < 0.05, two-way ANOVA: stress challenge x dietary treatment, Tukey post-hoc test). Values expressed in mean ± SD, *n* = 9 samples/diet/sampling point. Control (control diet), GMOS (GMOS diet, 5000 ppm galactomannan–oligosaccharides), PHYTO (PHYTO diet, 200 ppm mixture of garlic and labiate plant essential oils).

**Table 4 animals-12-03332-t004:** *Dicentrarchus labrax* gill relative gene-expression values at 2 h after confinement stress challenge (C challenge) or confinement combined with infection with the pathogen *Vibrio anguillarum* (CI challenge).

	Confinement (C Challenge)			Confinement + Infection (CI Challenge)		
	Control	GMOS	PHYTO	Control	GMOS	PHYTO
*nfΚβ2*	1.02 ^a^ ± 0.26	0.2 ^b^ ± 0.22	0.29 ^b^ ± 0.22	0.99 ^a^ ± 0.20	0.2 ^b^ ± 0.03	0.23 ^b^ ± 0.10
*hif-1α*	1 ^a^ ± 0.01	0.3 ^b^ ± 0.1	0.4 ^b^ ± 0.2	1 ^a^ ± 0.2	0.4 ^b^ ± 0.01	0.6 ^b^ ± 0.1
*gr*	1.01 ^a^ ± 0.17	0.21 ^b^ ± 0.2	0.18 ^b^ ± 0.13	0.94 ^a^ ± 0.25	0.21 ^b^ ± 0.02	0.24 ^b^ ± 0.03
*nd5*	1.06 ^a^* ± 0.50	0.19 ^b^ ± 0.07	0.30 ^b^ ± 0.19	0.37 ** ± 0.07	0.2 ± 0.02	0.46 ± 0.24
*coxi*	1 ^a^* ± 0.14	0.7 ^b^ ± 0.06	0.1 ^b^ ± 0.1	0.22 ** ± 0.09	0.1 ± 0.03	0.1 ± 0.07
*sod*	1.08 ^a^ ± 0.58	0.2 ^b^ ± 0.26	0.22 ^b^ ± 0.21	0.85 ^a^ ± 0.48	0.3 ^b^ ± 0.13	0.17 ^b^ ± 0.08
*cat*	1 ^a^ ± 0.09	0.19 ^b^ ± 0.16	0.26 ^b^ ± 0.2	0.73 ± 0.31	0.26 ± 0.17	0.35 ± 0.17
*gpx*	1 ± 0.13	0.94 ± 0.89	0.42 ± 0.18	3.57 ^a^ ± 2.36	0.42 ^b^ ± 0.26	0.62 ^b^ ± 0.26
*ocln*	1.01 ± 0.18	0.82 ± 0.74	0.8 ± 0.29	1.48 ± 0.75	0.78 ± 0.01	1.2 ± 0.38
*zo-1*	1.08 ± 0.56	2.18 ± 0.62	3.02 ± 0.88	2.64 ± 1.7	3.59 ± 1.32	2.94 ± 1.33
*hsp70*	1 ^a^ ± 0.09	0.11 ^b^ ± 0.08	0.19 ^b^ ± 0.13	0.56 ± 0.04	0.28 ± 0.2	0.16 ± 0.05
*hsp90*	1 ^a^ ± 0.01	0.2 ^b^ ± 0.2	0.2 ^b^ ± 0.2	1.3 ^a^ ± 0.8	0.2 ^b^ ± 0.01	0.2 ^b^ ± 0.2
*NKA α1a*	1 ^a^ ± 0.11	0.28 ^b^ ± 0.32	0.26 ^b^ ± 0.13	1.48 ^a^ ± 0.18	0.17 ^b^ ± 0.01	0.44 ^b^ ± 0.26

Different letters denote significant differences among dietary treatments at each stress challenge (*p* < 0.05, two-way ANOVA: stress challenge x dietary treatment, Tukey post-hoc test). Different numbers of asterisks (*) denote significant differences among C and CI challenge for each dietary treatment (*p* < 0.05, two-way ANOVA: stress challenge x dietary treatment, Tukey post-hoc test). Values expressed in mean ± SD, *n* = 3 samples/diet/sampling point. Control (control diet), GMOS (GMOS diet, 5000 ppm galactomannan–oligosaccharides), PHYTO (PHYTO diet, 200 ppm mixture of garlic and labiate plant essential oils). Target genes: *nfκβ2*: nuclear factor kappa beta-2, *hif-1α:* hypoxia-inducible factor 1 alpha, *gr*: glucocorticoid receptor, *nd5*: NADH dehydrogenase subunit 5, *coxi*: cytochrome c oxidase subunit 1, *sod*: superoxide dismutase, *cat*: catalase, *gpx*: glutathione peroxidase, *zo-1*: zonula occludens-1, *ocln*: occludin, *hsp70*: heat-shock protein 70, *hsp90*: heat-shock protein 90, *NKA α1a*: Na^+^/K^+^ ATPase subunit *α*1a, *α-tubulin* (housekeeping).

**Table 5 animals-12-03332-t005:** *Dicentrarchus labrax* gill relative gene-expression values at 24 h after confinement stress challenge (C challenge) or confinement combined with infection with the pathogen *Vibrio anguillarum* (CI challenge).

	Confinement (C Challenge)			Confinement + Infection (CI Challenge)		
	Control	GMOS	PHYTO	Control	GMOS	PHYTO
*nfΚβ2*	0.22 ^b^ ± 0.05	1.01^a^ ± 0.04	0.51^ab^ ± 0.31	0.73 ± 0.15	0.66 ± 0.24	0.78 ± 0.11
*hif-1a*	1 ± 0.3	2.22 ± 0.9	1.82 ± 0.9	1.2 ± 0.2	1.8 ± 1	1.6 ± 0.2
*gr*	0.30 ± 0.08	1.08 ± 0.35	0.64 ± 0.37	0.71 ± 0.19	0.82 ± 0.52	1.07 ± 0.34
*nd5*	0.21 ± 0.07	0.5 ± 0.19	0.48 ± 0.39	0.48 ± 0.2	0.52 ± 0.19	0.57 ± 0.29
*coxi*	0.12 ± 0.06	0.36 ± 0.14	0.33 ± 0.29	0.38 ± 0.10	0.45 ± 0.34	0.54 ± 0.4
*sod*	0.63 ^b^* ± 0.23	2.72 ^a^* ± 0.24	0.48 ^b^ ± 0.15	2.4 ** ± 0.03	1.1 ** ± 0.17	1.06 ± 0.17
*cat*	0.28 ± 0.17	1.53 ± 0.23	0.84 ± 0.63	1.64 ± 1.34	1.59 ± 0.99	1.38 ± 1.03
*gpx*	0.55 ^b^ ± 0.26	1.85 ^a^ ± 0.6	0.88 ^ab^ ± 0.57	1.24 ± 0.09	1.1 ± 0.49	1.51 ± 0.29
*ocln*	0.1 ^b^ ± 0.03	0.18 ^ab^ ± 0.11	0.49 ^a^ ± 0.3	0.24 ± 0.1	0.47 ± 0.07	0.38 ± 0.15
*zo-1*	5.04 * ± 1.69	3.35 * ± 0.93	4.33 * ± 1.04	3.24 ** ± 0.6	2.68 ** ± 1.21	2.14 ** ± 1.08
*hsp70*	0.12 * ± 0.04	0.32 ± 0.01	0.16 ± 0.04	0.63 ** ± 0.2	0.28 ± 0.13	0.51 ± 0.23
*hsp90*	1.1 ^b^ ± 0.5	4.12 ^a^ ± 1.3	2 ^ab^ ± 1.9	1.2 ± 0.4	1.5 ± 1.7	1.7 ± 0.4
*NKA α1a*	0.39 ± 0.24	1.03 ± 0.45	0.51 ± 0.08	0.85 ± 0.42	0.6 ± 0.22	1.1 ± 0.09

Different letters denote significant differences among dietary treatments at each stress challenge (*p* < 0.05, two-way ANOVA: stress challenge x dietary treatment, Tukey post-hoc test). Different numbers of asterisks (*) denote significant differences among C and CI challenge for each dietary treatment (*p* < 0.05, two-way ANOVA: stress challenge x dietary treatment, Tukey post-hoc test). Values expressed in mean ± SD, *n* = 3 samples/diet/sampling point. Control (control diet), GMOS (GMOS diet, 5000 ppm galactomannan–oligosaccharides), PHYTO (PHYTO diet, 200 ppm mixture of garlic and labiate plant essential oils). Target genes: *nfκβ2*: nuclear factor kappa beta-2, *hif-1α:* hypoxia-inducible factor 1 alpha, *gr*: glucocorticoid receptor, *nd5*: NADH dehydrogenase subunit 5, *coxi*: cytochrome c oxidase subunit 1, *sod*: superoxide dismutase, *cat*: catalase, *gpx*: glutathione peroxidase, *zo-1*: zonula occludens-1, *ocln*: occludin, *hsp70*: heat-shock protein 70, *hsp90*: heat-shock protein 90, *NKA α1a*: Na^+^/K^+^ ATPase subunit *α*1a, *α-tubulin* (housekeeping).

**Table 6 animals-12-03332-t006:** *Dicentrarchus labrax* gill relative gene-expression values at 168 h after confinement stress challenge (C challenge) or confinement combined with infection with the pathogen *Vibrio anguillarum* (CI challenge).

	Confinement (C Challenge)			Confinement + Infection (CI Challenge)		
	Control	GMOS	PHYTO	Control	GMOS	PHYTO
*nfΚβ2*	0.27 ± 0.17	0.47 ± 0.18	0.40 ± 0.08	0.55 ± 0.08	0.34 ± 0.11	0.43 ± 0.10
*hif-1a*	1 ± 0.3	1.3 ± 0.3	1 ± 0.3	1.1 ± 0.2	0.7 ± 0.3	0.7 ± 0.2
*gr*	0.24 ± 0.18	0.46 ± 0.09	0.6 ± 0.13	0.53 ^a^ ± 0.04	0.27 ^b^ ± 0.11	0.65 ^a^ ± 0.17
*nd5*	0.25 ± 0.16	0.26 ± 0.11	0.39 ± 0.11	0.35 ± 0.03	0.19 ± 0.06	0.46 ± 0.21
*coxi*	0.17 ± 0.16	0.17 ± 0.13	0.21 ± 0.07	0.28 ± 0.02	0.11 ± 0.06	0.29 ± 0.1
*sod*	0.31 * ± 0.2	0.47 ± 0.19	0.38 ± 0.19	1.6 ^a^** ± 0.56	0.31 ^b^ ± 0.17	0.54 ^b^ ± 0.13
*cat*	0.37 ± 0.33	0.54 ± 0.33	0.59 ± 0.12	1.11 ± 0.62	0.51 ± 0.10	0.68 ± 0.25
*gpx*	0.74 ± 0.22	1.17 ± 0.19	1.02 ± 0.11	0.96 ^b^ ± 0.25	0.46 ^b^ ± 0.03	1.39 ^a^ ± 0.38
*ocln*	0.68 ^a^* ± 0.09	0.51 ^ab^*± 0.19	0.20 ^b^ ± 0.02	0.17 ^b^** ± 0.02	0.89 ^a^** ± 0.14	0.43 ^ab^ ± 0.01
*zo-1*	2.47 ± 0.15	1.44 * ± 0.64	1.57 ± 0.66	2.62 ^ab^ ± 1.31	3.57 ^a^** ± 0.62	0.86 ^b^ ± 0.11
*hsp70*	0.12 * ± 0.05	0.17 ± 0.06	0.15 ± 0.06	0.47 ** ± 0.01	0.19 ± 0.05	0.26 ± 0.08
*hsp90*	2 ± 2.1	4.5 * ± 1.4	3.3 ± 0.7	1.1 ± 0.3	0.7 ** ± 0.4	1.2 ± 0.1
*NKA α1a*	0.2 ^b^* ± 0.1	0.51 ^a^ ± 0.04	0.37 ^ab^ ± 0.08	0.61 ** ± 0.11	0.64 ± 0.08	0.49 ± 0.1

Different letters denote significant differences among dietary treatments at each stress challenge (*p* < 0.05, two-way ANOVA: stress challenge x dietary treatment, Tukey post-hoc test). Different numbers of asterisks (*) denote significant differences among C and CI challenge for each dietary treatment (*p* < 0.05, two-way ANOVA: stress challenge x dietary treatment, Tukey post-hoc test). Values expressed in mean ± SD, *n* = 3 samples/diet/sampling point. Control (control diet), GMOS (GMOS diet, 5000 ppm galactomannan–oligosaccharides), PHYTO (PHYTO diet, 200 ppm mixture of garlic and labiate plant essential oils). Target genes: *nfκβ2*: nuclear factor kappa beta-2, *hif-1α:* hypoxia-inducible factor 1 alpha, *gr*: glucocorticoid receptor, *nd5*: NADH dehydrogenase subunit 5, *coxi*: cytochrome c oxidase subunit 1, *sod*: superoxide dismutase, *cat*: catalase, *gpx*: glutathione peroxidase, *zo-1*: zonula occludens-1, *ocln*: occludin, *hsp70*: heat-shock protein 70, *hsp90*: heat-shock protein 90, *NKA α1a*: Na^+^/K^+^ ATPase subunit *α*1a, *α-tubulin* (housekeeping).

## Data Availability

Not applicable.

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
