# Peer review of "Gill Oxidative Stress Protection through the Use of Phytogenics and Galactomannan Oligosaccharides as Functional Additives in Practical Diets for European Sea Bass (Dicentrarchus labrax) Juveniles"

_animals, 2022, doi:10.3390/ani12233332_

Round 1

Reviewer 1 Report

General comment:

The present work analyzed ameliorating effect of feed additives on gill’s oxidative stress in fish, under the conditions of both confinement and Vibrio anguillarum infection. The previously reported antioxidant ingredients were used to test. Well, the experimental design was not sufficiently described. Both background of the research and the explanation of the findings were not clearly provided. As to the qPCR technique, which was the important method used in this study, the major concern is that current used internal reference was not suitable. Additional experiments and demonstration of the underlying mechanism were required. Thus, the manuscript needs a major revision.

Specific comments:

Introduction:

1.     Line 112-114: Not all the plant-sourced ingredients are beneficial, yet some components were even called anti-nutrients, such as saponins.

2.     Line 116-118: The molecular mechanism for PFAs derived from garlic and labiate plants was not detailed.

3.     Line 118-119: So, GMOS is a kind of prebiotics? Detail the involved mechanism in the introduction.

4.     There lacks why paralleling the two functional additives (pytogenics and galactomannan oligosaccharides) together in this study.

Materials and methods:

1.     It is necessary to describe the sampling method in detail, using a separate section.

2.     The experimental setup should be illustrated by a diagram. See the paper entitled “Dietary supplementation of Bacillus velezensis improves Vibrio anguillarum clearance in European sea bass by activating essential innate immune mechanisms” published in Fish and Shellfish Immunology.

3.     Line 197: alpha-tubulin is not a stable internal reference gene for qPCR analysis during infection. According to the reference entitled “Virulence factors impair epithelial junctions during bacterial infection” published in Journal of Clinical Laboratory Analysis, the tubulin protein could interact with bacteria’s components. The better internal reference gene in qPCR analysis for fish infection could be found in the paper entitled “Evaluation of internal control genes for qRT-PCR normalization in tissues and cell culture for antiviral studies of grass carp (Ctenopharyngodon idella)” published in Fish and Shellfish Immunology.

Results:

1.     It is necessary to show the diagrams for both cumulative mortality and relative percent survival as new figures.

2.     EF1a and 18s are much suitable genes to normalize the qPCR result during infection in fish. Therefore, the qPCR result should be re-calculated, with additional expression analysis of stable internal reference genes.

3.     Fig. 1: Most region of the heatmap was without obvious changes. So, the 2h after challenge may be not the best time point to check these gene expression. Did the author do some pre-experiment for the time course, such as the qPCR, to determine the sampling time points?

4.     Line 353: Again, why did authors do sampling at this time point?

Discussion:

1.     The discussion should be written based on current findings. Yet, current structing of text in the discussion was not in a clear logic.

2.     There lacked a small paragraph to summarize the main findings at the beginning of the discussion.

3.     In Fig.2, for C treatment, gene expression in the GMOS group differed a lot to that in PHYTO group. Current discussion has not explained this point.

4.     What is the different mechanism for current used two additives to ameliorate the oxidative stress in the gill? This is not clearly explained in the discussion.

5.     Then, a diagram of the hypothetical mechanism would be drawn according to the result and discussion. It is then much better to put forward the concept.

Other comments:

1.     The manuscript should be revised by an English native speaker. Some gramma mistakes still could be found in the text.

2.     All “p”, represented p-value, should be italic.

3.     For all mathematical symbols, there should be a space either before or after the symbol. Please check these points throughout the text.

Author Response

Authors would like to thank you for the suggested changes and improvements to the manuscript, which have helped to substantially improve the quality of the manuscript. All comments/suggestions have been included/adapted along the text. Please find attached PDF file with Reviewer 1 answers. Author's answers and comments are in blue fonds. Some schemes and figures have been included in response to reviewer 1. 

Reviewer 2 Report

The manuscript by Montero et al. evaluated the use of functional additives in diets for European seabass juveniles and how their supplementation alleviates the gene expression of stress markers in their gills. Overall, the manuscript is well-written and apparently properly conducted. There are a few minor mistakes that the authors should pay attention to throughout the manuscript.

Please find below some remarks and comments.

L21-24: This sentence is convoluted. Please restructure it.

L39-40: I suggest changing the challenge abbreviations to “C challenge” and “CI challenge” instead of treatments.

L40-41: Please refrain from presenting P-values in the abstract.

L179: Correct to ammonium sulphate.

L186: Unit probably missing from total mRNA.

L231: Missing words between 158 and 16.3%.

L236: It was previously stated that the criteria for significance was P<0.05. I recommend deleting all the repeated P<0.05 throughout the result section.

Table 1: Control and GMOS diets sum up to 99.96, and PHYTO sums up to 99.98. Please double-check your formulation.

Table 2: P-values should be presented for the main factors and the interaction between the main factors.

Author Response

We would like to thank the reviewer 2 for the suggested changes, which have been adapted along the text, substantially improving the manuscript quality

Introduction

  • Line 22, please be specific with phytogenic feed additives; it is too general.

The present study evaluates the effects of the plant origin galactomannan-oligosaccharides (GMOS) and a mixture of garlic and labiate plants essential oils (PHYTO) as potential boosters of European sea bass (Dicentrarchus labrax) juveniles fed low FM/FO based diets gill endogenous antioxidant capacity against physical and biological stressors.

  • Line 40, please be specific as in the abstract you should clearly mention what the name of the treatment is.

Both, GMOS and PHYTO diets attenuated fish stress response inducing lower (P < 0.05) circulating plasma cortisol after the stress challenge

  • Please mention the novelty of your work in the last paragraph of the introduction.

A scarce number of studies have investigated the effects of functional ingredients to offset the negative effects derived from low FM/FO diet formulation and specially in fish subjected to acute stress processes. Thus, the aim of this study is to evaluate the effects of functional additives (PFAs or GMOS) as potential boosters of European sea bass (Dicentrarchus labrax) juveniles fed low fish meal (FM)/fish oil (FO) based diets gill endogenous antioxidant capacity when challenge against physical and biological stressors.

Materials and Methods

  • Please explain a little bit about your experimental diets, per each Table and Figure. Each Table and figure should represent enough information separately from the text.
  •  

The following information has been added when necessary

  • Control (control diet), GMOS (GMOS diet, 5000 ppm galactomannan-oligogaccharides), PHYTO (PHYTO diet, 200 ppm mixture of garlic and labiate plants essential oils).
  • α-tubulin (housekeeping).
  • South-American, Superprime 68%.

2 South American fish oil.

3 DLG AS, Denmark.

4 Vilomix, Denmark.

5 BAROX BECP, Ethoxyquin.

6Delacon Biotechnik GmbH, Austria.

7Delacon Biotechnik GmbH, Austria.

Reviewer 3 Report

The authors investigated the effect of gill oxidative stress protection through the use of phytogenics and galactomannan oligosaccharides as functional additives in practical diets for European sea bass (dicentrarchus labrax) juveniles. They designed six treatments to test their hypothesis. This manuscript (MS) was clearly written and easy to understand. This work can help the sustainability of this species farming as few studies have been done on this topic. However, some minor issues significantly compromised the quality of this MS.

Minor comments

·       Line 22, please be specific with phytogenic feed additives; it is too general.

·       Line 40, please be specific as in the abstract you should clearly mention what the name of the treatment is.

·       Line 89-119, please only focus on the parameters that you measured in this study.

·       Throughout the MS, please first mention the common name plus scientific name, and for the rest of the MS, just report the common name.

·       Here and elsewhere, report P uppercase and italic (P<0.05).

·       Throughout the MS, if there is no significant difference, no need to report P-value.

·       Please reorder the keywords alphabetically and capitalize each word.

·       Please update the introduction with recent works as many studies are available from the last two years, which were not included in this section.

·       Please mention the novelty of your work in the last paragraph of the introduction.

·       Although you wrote the discussion well, you can still improve it by answering these questions and annotating them in the discussion section. Why were these results observed? Discuss more possible reasons.

Tables and Figures

•            Please explain a little bit about your experimental diets, per each Table and Figure. Each Table and figure should represent enough information separately from the text.

•            Double-check the units and titles of all Tables.

•            Please mention in the footnote of all Tables which kind of statistical method you used for comparing the means.

When revising your manuscript, please consider all issues mentioned in the reviewers' comments carefully with clear outlines for every change made in response to their comments including suitable rebuttals for any comments you deem inappropriate. Please itemize your response to each review comment and highlight the revised at re-submission.

Best regards

Author Response

First of all, we have to thank the reviewer suggestions. All the commentaries have been included/adapted in the text, improving the manuscript quality

  • This sentence is convoluted. Please restructure it.

The present study evaluates the effects of two different functional ingredients, the plant origin galactomannan-oligosaccharides (GMOS) or a mixture of garlic and labiate plants essential oils (PHYTO), as potential boosters of gill endogenous antioxidant capacity in European sea bass (Dicentrarchus labrax) juveniles fed low FM/FO based diets

  • I suggest the challenge abbreviations to “C challenge” and “CI challenge”

The reviewer’s suggestion has been accepted and changed along the manuscript

  • Please refrain from presenting P-values in the abstract.

Changed

  • Correct to ammonium sulphate.

Changed

  • Unit probably missing from total mRNA

(ng/uL). corrected. 

  • It was previously stated that the criteria for significance was P<0.05. I recommend deleting all the repeated P<0.05 throughout the result section.

Changed

  • Table 1: Control and GMOS diets sum up to 99.96, and PHYTO sums up to 99.98. Please double-check your formulation.

We would like to thanks reviewer 3 to detect this mistake. This is due to the automatic rounding of the tables in Excel. The data have been accordingly adjusted.

  • L231: Missing words between 158 and 16.3%.

Corrected

  • Table 2: P-values should be presented for the main factors and the interaction between the main factors.

Added in Table: Different letters denote significant differences among dietary treatments at each stress treatment (P < 0.05, Two-way ANOVA: Stress treatment x Dietary treatment, Tukey post-hoc test). Different number of asterisk symbols (*) denotes significant differences among C and CI treatment for each dietary treatment (P < 0.05, Two-way ANOVA: Stress treatment x Dietary treatment, Tukey post-hoc test).

Round 2

Reviewer 1 Report

The revision has improved a lot. The experimental design and the result for survival have been illustrated using new Figure 1 and 2, respectively. Well, before acceptance, there are still several points to be solved, as following: 1. In line 245-246: eEF1ɑ1 has been added in the Table 2. I think it was meant for recalculating qPCR result. But in the text, the internal reference was still ɑ-tublin. So, how about the eEF1ɑ? As previous commented, the internal reference should be replaced with more suitable ones, such as EF1ɑ, especially in the inflammatory condition for the CI challenge. 2. The title text in the Figure 2 is not clear. The diagram with higher resolution is required. 3. There is still some “p” represented for p value not in italic, as in line 275 and 277. 4. In the abbreviation list, for “FW (Fordward primer sequence)”, the font size is not in consistency with other abbreviations.

Author Response

Authors would like to thanks again reviewer 1 for the detailed revision of the manuscript to improve the quality.

RW: The revision has improved a lot. The experimental design and the result for survival have been illustrated using new Figure 1 and 2, respectively. Well, before acceptance, there are still several points to be solved, as following: 1. In line 245-246: eEF1ɑ1 has been added in the Table 2. I think it was meant for recalculating qPCR result. But in the text, the internal reference was still ɑ-tublin. So, how about the eEF1ɑ? As previous commented, the internal reference should be replaced with more suitable ones, such as EF1ɑ, especially in the inflammatory condition for the CI challenge.

AU: a paragraph has been included to clarify for readers how the housekeeping has been chosen in accordance with the experimental design and the CFX MaestroTM Software selection tool. The primer sequences for all the housekeeping genes used in the present work have been updated in the list. 

RW: The title text in the Figure 2 is not clear. The diagram with higher resolution is required. 3. There is still some “p” represented for p value not in italic, as in line 275 and 277. 4. In the abbreviation list, for “FW (Fordward primer sequence)”, the font size is not in consistency with other abbreviations. 

AU: Title text in the figure 2 has been improved (in yellow) and resolution of the diagram has been also improved. P values has been checked and the font go FW is now with the similar font type that the rest of the abbreviations.